# No Evidence of the Vertical Transmission of Non-Virulent Infectious Salmon Anaemia Virus (ISAV-HPR0) in Farmed Atlantic Salmon

**DOI:** 10.3390/v13122428

**Published:** 2021-12-03

**Authors:** Debes Hammershaimb Christiansen, Petra Elisabeth Petersen, Maria Marjunardóttir Dahl, Nicolina Vest, Maria Aamelfot, Anja Bråthen Kristoffersen, Mona Dverdal Jansen, Iveta Matejusova, Michael D. Gallagher, Gísli Jónsson, Eduardo Rodriguez, Johanna Hol Fosse, Knut Falk

**Affiliations:** 1National Reference Laboratory for Fish and Animal Diseases, Faroese Food and Veterinary Authority, 110 Torshavn, Faroe Islands; pep@hfs.fo (P.E.P.); mmd@hfs.fo (M.M.D.); nv@hfs.fo (N.V.); 2Norwegian Veterinary Institute, 0454 Oslo, Norway; Maria.Aamelfot@fhi.no (M.A.); anjabrathen.kristoffersen@fhi.no (A.B.K.); mona-dverdal.jansen@vetinst.no (M.D.J.); Johanna.Hol.Fosse@vetinst.no (J.H.F.); knut.falk@gmail.com (K.F.); 3Marine Scotland Science, Marine Laboratory, Aberdeen AB11 9DB, UK; Iveta.Matejusova@gov.scot; 4The Roslin Institute and Royal (Dick) School of Veterinary Studies, University of Edinburgh, Edinburgh EH8 9YL, UK; mike.d.gallagher.16@gmail.com; 5Icelandic Food and Veterinary Authority, 220 Hafnarfjordur, Iceland; gisli.jonsson@mast.is; 6Benchmark Genetics Iceland, 220 Hafnarfjordur, Iceland; eduardo.rodriguez@bmkgenetics.com

**Keywords:** aquaculture, orthomyxoviridae, isavirus, ISAV-HPR0, transmission pathways, epidemiology, house-strains, viral fitness

## Abstract

The nonvirulent infectious salmon anaemia virus (ISAV-HPR0) is the putative progenitor for virulent-ISAV, and a potential risk factor for the development of infectious salmon anaemia (ISA). Understanding the transmission dynamics of ISAV-HPR0 is fundamental to proper management and mitigation strategies. Here, we demonstrate that ISAV-HPR0 causes prevalent and transient infections in all three production stages of Atlantic salmon in the Faroe Islands. Phylogenetic analysis of the haemagglutinin-esterase gene from 247 salmon showed a clear geographical structuring into two significantly distinct HPR0-subgroups, which were designated G2 and G4. Whereas G2 and G4 co-circulated in marine farms, Faroese broodfish were predominantly infected by G2, and smolt were predominantly infected by G4. This infection pattern was confirmed by our G2- and G4-specific RT-qPCR assays. Moreover, the HPR0 variants detected in Icelandic and Norwegian broodfish were never detected in the Faroe Islands, despite the extensive import of ova from both countries. Accordingly, the vertical transmission of HPR0 from broodfish to progeny is uncommon. Phylogenetic and statistical analysis suggest that HPR0 persists in the smolt farms as “house-strains”, and that new HPR0 variants are occasionally introduced from the marine environment, probably by HPR0-contaminated sea-spray. Thus, high biosecurity—including water and air intake—is required to avoid the introduction of pathogens to the smolt farms.

## 1. Introduction

Infectious salmon anaemia (ISA) is an important disease of farmed Atlantic salmon, *Salmo salar* L. Outbreaks of ISA were first identified in Norway in 1984 [1]; New Brunswick, Canada in 1996 [2]; Scotland in 1998 [3]; the Faroe Islands in 2000 [4]; Maine, USA in 2001 [5]; and Chile in 2007 [6]. ISA is caused by the infectious salmon anaemia virus (ISAV) belonging to the *Orthomyxoviridae* family [7]. The viral genome is composed of eight single-stranded RNA segments with negative polarity, encoding for at least 10 proteins, including two surface glycoproteins: the fusion protein (F) encoded by segment 5 and the haemagglutinin-esterase protein (HE) encoded by segment 6 [8].

ISAV exist in two phenotypically distinct variants, the virulent ISAV-HPRΔ (HPRΔ) and nonvirulent ISAV-HPR0 (HPR0) [4,8], which are distinguished by mutations in the *HE* and *F* genes. All of the known virulent HPRΔ variants carry various deletions in the highly polymorphic region (HPR) of the *HE*-gene [9,10,11] and a Q_266_L substitution or an insertion adjacent to the cleavage site of the *F*-gene [12,13]. Virulent ISAV strains arise from mutations, deletions and insertions within the *F* and *HE* genes of HPR0 [14], probably because of selective pressure associated with the intensive Atlantic salmon farming environment. Thus, HPR0 is considered the wild-type strain of ISAV.

All of the variants of the HPRΔ and HPR0 strains can be divided into two major phylogenetic genogroups: the North American and the European [11,15,16]. The European genogroup has been further subdivided into four subgroups, here designated subgroup 1 (G1), subgroup 2 (G2), subgroup 3 (G3) and subgroup 4 (G4) (former EU-G1 (Clade III), EU-G2 (Clade II), EU-G3 (Clade IV) and EU-NA (Clade I), respectively) [4,11,14,15,17]. Phylogenetic analysis of the *F* gene has supported this classification [18].

Although HPRΔ has been demonstrated in all three production stages of Atlantic salmon, i.e., broodfish, smolt farms and marine sites in Norway [16,17,19], most of the ISA outbreaks in Norway and all of the outbreaks in the Faroe Islands occurred at marine farming sites. HPRΔ is transmitted horizontally between fish, between cages within a marine farming site, and between marine farming sites [19,20]. Accordingly, the recommended control measures rely on rapid containment and the eradication of the infected populations [21,22,23].

HPR0 has been detected in wild anadromous Atlantic salmon in Norway [16,17,24], Scotland [9] and Chile [25]. Because new HPRΔ variants may arise from HPR0 [14,26], its common occurrence and widespread distribution in farmed [4,14,16,17,26] and wild [17,24] Atlantic salmon makes it a potential risk factor for the emergence of new HPRΔ and ISA disease. Hence, understanding HPR0’s transmission mode and route is fundamental for proper management and mitigation strategies in Atlantic salmon farming. While HPR0 has also been found in each of the three production stages of Atlantic salmon [16,17,27], very little is known about HPR0 transmission dynamics between the production stages. Several studies have suggested that HPR0 is mainly transmitted vertically from broodfish to progeny [16,17,25]. However, because HPR0 cannot be cultured in cell-lines, no experimental evidence of the vertical transmission of HPR0 has been demonstrated. Studies of field material have suggested that horizontal spread is the major transmission route for HPR0 in the marine environment [26], including the confinement of the infection to the gills and skin [28], the time lag between sea-water transfer and the first detection of HPR0-infection, and the transience and seasonality of HPR0-infection [4].

In order to address the knowledge gap regarding HPR0 transmission dynamics, we genetically and epidemiologically characterized a comprehensive panel of HPR0 variants from all of the production stages in the Faroe Islands. Furthermore, we included HPR0 variants from Icelandic and Norwegian broodfish delivering fertilized ova to Faroese smolt farms. Our findings demonstrate that HPR0 is mainly transmitted horizontally between the three production stages of Atlantic salmon in the Faroe Islands.

## 2. Materials and Methods

### 2.1. Organisation of the Faroese Farming Industry

Following the devastating ISA epidemic (2000 to 2005) in the Faroe Islands [4], the production of Atlantic salmon has been organized into three different production stages; one land-based indoor broodfish farm with the production of ova and juveniles, eight land-based indoor hatcheries and pre-smolt production farms (hereafter referred to as smolt farms)—all of them using re-circulatory aquatic systems (RAS)—and 25 marine sites for the final growth phase in the sea (epidemiological zones; hereafter referred to as marine farms) with a minimum separation distance of 5 km**.** Between 1986 and 2002, the Faroese farming industry was supplied with ova exclusively from the Faroese broodfish farm. However, a change in the Faroese legislation in 2002 allowed the importation of ova. As a result, the industry has been supplied with ova from Faroese, Icelandic and Norwegian broodfish since 2003. According to Faroese legislation on Atlantic salmon aquaculture, all of the eggs must be disinfected at least twice with Buffodine^®^, as recommended by the manufacturer (Evans Vanodine International, Preston, UK). First, as newly stripped eggs after fertilization, and second as eyed eggs upon arrival in the smolt farms.

### 2.2. Study Period and Fish Cohorts

The surveillance and sampling of gills for infection with ISAV by real-time RT-PCR was initiated in Faroese marine farms in 2005, in the Faroese broodfish and smolt farms in 2007, and in the two Icelandic broodfish farms in 2009. Therefore, the study period spanned eight years (2007 to 2014) for the Faroese broodfish-, smolt- and marine farms and six years (2009 to 2014) for the Icelandic broodfish farms. The study population included the Faroese and the two Icelandic broodfish farms (broodfish farms I, II and III (BI–BII)), six Faroese smolt farms (smolt farms I–VI (SI–SVI)) with production of Atlantic salmon smolts the whole study period, and all 25 Faroese marine production sites (Marine farms I–XXV (MI–MXXV) (Appendix A). Two Faroese smolt farms were excluded from the study. One farm switched from the production of Rainbow trout to Atlantic salmon smolt in 2010, and the other one started the production of Atlantic salmon smolt in 2011. Both the Faroese and the Icelandic broodfish farms were land-based indoor facilities close to the shore. The Faroese broodfish farm had two sites on different islands: one freshwater flow-through site for egg stripping, hatching and smolt production, and another grow-out site for new generations of broodfish. This was a flow-through site supplied with seawater from 18 m depth that was filtered and UV-treated before entering the tanks. The two Icelandic broodfish farms were located in different geographic regions/watersheds, and were supplied with sea- and freshwater from different boreholes that were naturally filtered through porous lava. Both Icelandic broodfish farms were supplied with eggs and smolt exclusively from the same land-based indoor smolt farm.

The six Faroese smolt farms were located in different geographic regions/watersheds and used only freshwater throughout the production period, which was filtered and UV-treated before entering the farms. Some of the smolt farms used flow-through spring water (also filtered and UV-treated) at the start of feeding. All of the RAS systems were supplied with freshwater from rivers upstream of the smolt farms. All but one smolt of the farms (SII) were equipped with their own hatcheries. SII was supplied with fry either directly from the Faroese broodfish farm or from smolt farm SI.

A fish cohort was defined as the fish stock at a farming site at the time of sampling. Fish cohorts at a given farming site separated by a fallowing period were counted as different fish cohorts [26,29].

Throughout the study period, a total of 140 million viable ova/fry were delivered from the Faroese (55%), Icelandic (30%) and Norwegian broodfish farms (15%) to the six smolt farms included in the present study (Appendix A).

### 2.3. Sample Collection

The Faroese broodfish were sampled and screened for the presence of ISAV at stripping from September to January each season. The Icelandic broodfish were sampled and screened for ISAV at stripping throughout the year from 2009 to 2014. The six Faroese smolt farms were sampled and screened 2–4 times per year in 2007, 2008, 2012, 2013 and 2014, and 8–10 times per year in 2009, 2010 and 2011. The Faroese marine farms were sampled and screened 5–12 times per year from 2007 to 2011, and 4 times per year from 2012 to 2014 [4]. All of the samplings were performed by certified personnel (veterinarians and fish health biologists) in accordance with Faroese regulations on animal health and welfare (Law about animalwelfare. Available online: http://www.logir.fo/Logtingslog/49-fra-30-04-2018-um-djoravaelferd-Djoravaelferdarlogin (accessed on 30 April 2018). Briefly, anaesthesia was induced by Finquel Vet (MSD Animal Health, Kenilworth, NJ, USA), and the fish was killed by a blow to the head before sampling.

### 2.4. Detection of ISAV RNA by RT-qPCR

Gill samples were collected in RLT lysis buffer, followed by RNA extraction and duplex one-step RT-qPCR for ISAV screening, as previously described [4,14]. The primers (LGC Biosearch Technologies, Aarhus, Denmark) and TaqMan^®^ probes (Applied Biosystems, Waltman, MA, USA) for ISAV segment 8 and Atlantic salmon elongation factor-1α have been described previously [14].

### 2.5. Sanger Sequencing of the HE Gene

The full-length ISAV segment 6 from a total of 247 samples from the three production stages (Appendix A) was amplified by one-step RT-PCR, and was subjected to Sanger sequencing to identify the segment 6 genotype, as previously outlined [4,14]. In samples with Ct-values below 26, the whole segment 6 was amplified using the following forward/reverse primers: 5′-GCAAAGATGGCACGATTCATAAT-3/5′–CGTACAACATCAAGAACGTCTTC-3′, generating a PCR fragment of 1372 nt. In samples with Ct-values above 26, the segment 6 was amplified as two overlapping segments, using the following forward/reverse primers: 5′-GCAAAGATGGCACGATTCATAAT–3/5′-CTCATCTARCTCAACGTTCCTCATG-3′ and 5′-GTGTCAGACACCTT–GAAGTGAG-3′/5′-CGTACAACATCAAGAACGTCTTC-3′, generating two PCR fragments of 739 and 728 nt, respectively. The RT-PCR amplicons were subjected to Sanger sequencing as previously outlined [4,14].

### 2.6. Illumina Sequencing of the HE Gene

For the next-generation sequencing of the ISAV *HE* gene, 22 selected samples (Appendix A) were reverse-transcribed with the Superscript IV kit including random hexamer primers according to the manufacturer instructions (New England Biolab, Ipswich, MA USA). Q5 Hot Start High Fidelity polymerase (New England Biolab) was used for the multiplex PCR amplification of the cDNA, including segment 6 primers designed in-house with Primal Scheme [30]. Before sequencing, the amplicons were purified and quantified, followed by library preparation with barcode and adapter ligation as outlined in Quick et al. The sequencing was performed on the Illumina MiSeq platform with the use of the MiSeq Reagent Kit v3 (660 cycle) according to the manufacturer instructions (Illumina, San Diego CA, USA). The sequences were analysed with the CLC Genomic Workbench 12.0 and Genomics (Qiagen, Hilden, Germany) [31].

### 2.7. Specific Detection of the G2 and G4 Genotypes by Real-rTime RT-PCR

Due to the low analytical sensitivity of Sanger sequencing, this technique cannot exclude the presence of minority variants of G2 in samples dominated by G4, or vice-versa. Therefore, we designed two highly sensitive genotype-specific real-time RT-PCR assays, each selectively targeting G2 or G4. In order to ensure that the two assays were not cross-reactive, we took advantage of five specific single nucleotide polymorphisms (SNP) (four SNP’s in the forward primer and one SNP in the reverse primer) that differed between G2 and G4. For the specific amplification of the G2 subgroup, we used the G2-specific forward primer 5′-CAAGTAGAGCAGCCTGCGAAT-3′ and the G2-specific reverse primer 5′- GCTGCAATCCAAATACATGCT-3′. For the specific amplification of the G4 subgroup, we used the G4-specific forward primer 5′-CCAAGTAGAGCAACCTTCGACG-3′ and the G4-specific reverse primer 5′-CTGCAATCCAAATACATGC**C**-3′. A general segment 6-specific MGB TaqMan probe, FAM-5′-CAGGTTTTGGGATTGCT-3′-MGB-NFQ, was used in both assays. The specificity of the primers and probe was tested by nucleotide BLAST (BLAST. Available online: https://blast.ncbi.nlm.nih.gov/Blast.cgi accessed on 1 December 2021) against all of the publicly available ISAV segment 6 sequences. The QuantiTect Probe One-Step RT-PCR reaction mixture and PCR amplification conditions were used, as outlined previously [4,14], on a QuantStudio 5 real-time PCR System (Applied Biosystems). The amplicons were analyzed with the QuantStudio™ Design and Analysis Software v1.5.1. (Applied Biosystems). In order to monitor the efficiency and to determine the relative limit of detection (LOD) of the G2- and G4 specific assays, two 10-fold serial dilutions of ISAV-HPR0 positive RNA, one of the G2-subtype, and another of the G4-subtype were diluted in ISAV-negative Atlantic salmon total RNA [4,14,32]. Controls were included in each run, as outlined previously [4]. The G4-specific assay demonstrated no amplification on samples with a high HPR0 viral load (Ct < 16) of the G2-subtype (Appendix A). Likewise, the G2-specific assay showed no amplification on samples with a high HPR0 viral load (Ct < 16) of the G4-subtype (Appendix A). Whereas the RT-PCR amplification and Sanger sequencing of HPR variants demonstrated an LOD of approximately 5^−1^–10^−1^, both the G2 and G4-specific RT-qPCR assay demonstrated a significantly lower LOD of approximately 10^−6^–10^−7^ using G2 and G4 HPR0 serial dilution, respectively (data not shown). Furthermore, the G2- and G4-specific assays showed high sensitivity for ISAV-HPR0 detection, comparable to the duplex ISAV RT-qPCR assay [4,14] on samples infected with either the G2 (Appendix A) or G4 (Appendix A) HPR0 subtypes.

### 2.8. Molecular Phylogenetic Analysis of the Segment 6 HE Gene

ISAV-HPR0 segment 6 sequences were aligned to publicly available ISAV segment 6 sequences using CLC Main Workbench 8.1. (Qiagen) Identical sequences were identified, and only a single representative sequence was retained in the dataset in order to reduce subsequent analytical bias. The final alignment consisted of 48 unique sequences spanning a region of 1153 nts of the *HE* gene, including positions 61 to 1214 with respect to the ISAV segment 6 FO/07/12 (ass. no. KX823921). This sequence stretch includes segments encoding for the highly polymorphic region (HPR), the transmembrane region, the cytoplasmic tail region, and the partial 3′ untranslated region. The phylogenetic relationship among the ISAV isolates was inferred using a maximum-likelihood-based approach implemented within the CLC Main Workbench 8.1. The sequences can be accessed in GenBank (Appendix A).

### 2.9. Statistical Analyses

A Wilcoxon rank sum test was used to compare the prevalence of HPR0 infection between the hatchery groups (low and high overall HPR0 prevalence). The sum of the ranked values was compared to the sum of random ranks to test for significant differences between the two groups. A dataset containing the paired nucleotide sequence difference between all of the combinations of 23 hatchery sequences was generated. One hatchery sample (SV FO/156-1/08) was excluded from the dataset because it belonged to the G2 genogroup, while the remainder of the hatchery sequences belonged to the G4 genogroup. A Wilcoxon rank sum test was used to test whether the nucleotide differences in sequences within hatcheries were significantly different to those between hatcheries. Based on sampling dates, the distances between hatchery sequences in the number of days were calculated, and variables that indicated whether the sequences originated from the same or different hatcheries were generated. The nucleotide distances were modelled by Poisson regression, using origin (same or different hatchery) and distance (days between sampling points) as explanatory variables. Based on this model, the predicted differences in the hatchery sequences over 7 years were additionally calculated. The data management and analyses were performed using R, version 4.0.5. (Title: R Core Team, 2020, Available on http://www.R-project.org (released 31 March 2021).

## 3. Results

### 3.1. HPR0 Infection Is Prevalent in Each of the Three Atlantic Salmon Production Stages

Throughout the eight-year study period from 2007 to 2014, we demonstrated the presence of ISAV in each of the three Faroese Atlantic salmon production stages (i.e., land-based broodfish, freshwater smolt and marine production fish) with an overall total prevalence of 8% (Table 1). The annual prevalence in the three production stages ranged from 0% to 93% in stripped broodfish, from 0% to 18% in smolt farms, and from 3% to 15% in marine farms. In Iceland, ISAV was detected for the first time in the two broodfish farms in 2009 (Table 1). The annual prevalence of ISAV ranged from 0.2% to 19.2% (Table 1). The highest prevalence of ISAV was detected in 2009 in outdoor tanks at the two broodfish farms. Consequently, all of the outdoor tanks were reconstructed as indoor tanks in 2010/2011. In the following years, prevalence of ISAV started to decline in the two Icelandic broodfish farms (Table 1). Despite the high prevalence of ISAV in all of the production stages, intensive surveillance revealed no clinical or gross pathological signs in HPR0-infected fish, or signs indicative of ISA disease [33]. Accordingly, the sequencing or fragment analysis of the HPR-region from more than 2000 samples from the three production stages revealed the non-virulent HPR0 variant in each of the cases (data not shown), with the exception of one ISAV-HPRΔ isolate (FO/121/14) detected at a marine farming site in 2014 [14].

### 3.2. The Prevalence of HPR0 Infection Varies within and between Smolt Farms

The overall prevalence of HPR0 varied significantly between the individual smolt-farms in the study (Table 2). Whereas smolt farms SI, SIII and SIV (the low-prevalence group) demonstrated an overall low annual prevalence ranging from 0% to 3%, the annual prevalence in smolt farms SII, SV and SVI (the high-prevalence group) was significantly higher, ranging from 16% to 19% (*p* < 0.001) (Table 2). Notably, we observed a marked variation in the prevalence of ISAV detection at different sampling points on the same farm, both within and between years, ranging from 0% to 100% (Appendix A). Given the large number of samples collected, this illustrates the challenges associated with detection of HPR0 to determine its presence or freedom of infection.

### 3.3. HPR0 Causes a Transient and Subclinical Infection in Each of the Three Production Stages

The patterns of HPR0 infection in the Faroese broodfish varied throughout the stripping periods in 2008, 2010 and 2011 (Figure 1). Whereas both the HPR0 prevalence and viral load increased substantially throughout the one-month stripping period in 2008 (Figure 1a,b), the viral prevalence was high (>90%) throughout the stripping period in 2010, but with a marked increase in viral load (Figure 1c,d). These data suggest the rapid dissemination of the virus within the brood stock population. In 2011, HPR0 was not detected before 76 days after the first stripping (Figure 1e,f). In 2009, 2012 and 2013, no HPR0 was detected in broodfish during the stripping periods.

A similar transient infection pattern was observed in the smolt farms infected with HPR0 (Figure 2, Appendix A). After the initial detection of the virus, HPR0 spread rapidly throughout the populations, with a transient peak prevalence of up to 100% in several cases (Figure 2a,c,e,g,i). The highest viral loads coincided with the peak prevalence (Figure 2b,d,f,g,j), in line with previous observations from marine sites [4].

### 3.4. HPR0 Prevalence and Load Is Negligible in the Spawn Fluid of HPR0-Positive Broodfish

From the stripping in 2010, both gills and spawn fluid from 82 broodfish were tested for the presence of ISAV. Whereas HPR0 was detected in 98% (80/82 fish) of the gill samples, only 12% (10/80 fish) of the spawn fluid samples were positive (Figure 3a). Similarly, the viral load in the gills was high (median Ct = 22), while the viral load in the spawn fluid was very low (median Ct = 35) and on the detection limit of the real-time RT-PCR assay (Figure 3b). Given that HPR0 appears to be an infection of mucosal epithelium [28], the high viral load in the gills compared to spawn fluid may suggest the cross-contamination of the spawn fluid during sampling. Nevertheless, appropriate disinfection is important to eliminate HPR0 contamination on the ova and subsequent transfer to progeny.

### 3.5. All of the Faroese HPR0 Variants Cluster in Subgroups G2 or G4

The alignment and genetic analysis of nucleotides 61 to 1214 (including the highly polymorphic region) of the ISAV *HE* gene open reading frame derived from 247 individual Faroese Atlantic salmon representing each of the three production stages (Appendix A) revealed that 34 unique HPR0 variants were detected in the study period, of which 14 clustered in subgroup G2 and 20 in subgroup G4 (Appendix A).

### 3.6. HPR0 Variants in Faroese, Icelandic and Norwegian Broodfish Are Geographically Structured

Throughout the study period, a total of 140 million ova were delivered from Faroese (55%), Icelandic (30%) and Norwegian (15%) broodfish to the six Faroese smolt farms (Appendix A). The phylogenetic relationships between the HPR0 variants from Norwegian, Icelandic and Faroese broodfish, as well as from the five HPR0-positive smolt farms (Table 2), are depicted in Figure 4. A total of 18 unique HPR0 variants were identified among the 61 individual HPR0-positive broodfish (31 Faroese broodfish, 17 Icelandic broodfish, and publicly available sequences from 13 Norwegian broodfish) (Figure 4). All of the Norwegian broodfish variants clustered within subgroups G1 or G3, all of the Icelandic variants clustered in G2, and all of the Faroese variants clustered in G2 or G4 (Figure 4). Whereas Faroese broodfish stripped in 2008 were co-infected with HPR0 in subgroups G2c and G4a, broodfish stripped in 2010 and 2011 were exclusively infected with HPR0 within subgroup G2d (Appendix A and Figure 4). This was confirmed with our G2- and G4-specific assays on 195 individual Faroese broodfish, where only G2 was detected in broodfish stripped in 2010 and 2012 (Figure 5a). As expected, all of the 56 tested broodfish stripped in 2008 disclosed the co-infection of G2 and G4 by the G2- and G4-specific assays (Figure 5a). It is noteworthy that the two Icelandic broodfish farming sites BI and BII were infected with two highly distinct HPR0 variant populations—G2a and G2b (Figure 4)—even though both farms were supplied with smolt originating from the same hatchery. Interestingly, the Faroese and Icelandic broodfish HPR0 variant populations in G2 were significantly different (Figure 4).

### 3.7. HPR0 Variants in the Faroese Smolt Farm Cluster in Subgroup G4

The phylogenetic analysis of ISAV-HPR0 detected between 2008 and 2014 in the five HPR0-positive smolt farms (Table 2 and Appendix A) identified 15 different HPR0 variants among 55 individual HPR0-positive smolts from the 21 HPR0-positive smolt cohorts (Figure 4). It is noteworthy that all but one of these 15 variants clustered in subgroup G4 (Figure 4). The only HPR0 variant in G2 was detected in smolt farm SV in April 2008, which at this stage was co-infected with HPR0 in G2c and G4a. All of the HPR0 variants detected by Sanger sequencing between 2009 and 2014, in 51 individual smolts from the five positive farms, clustered exclusively in G4 (Figure 4). Our G2- and G4-specific assays confirmed the exclusive presence of G4 in 213 individual smolts tested between 2009 and 2014 (Figure 5b). The only exception was a co-infection with G2 and G4 in 21 smolts tested in 2008 from smolt farm SV (Figure 5b). This observation was confirmed with our NGS analysis on samples from 20 selected smolts (Appendix A).

Taken together, these data demonstrate that there is little or no genetic link between the HPR0 subtypes (G1, G2 and G3) circulating in Faroese, Icelandic and Norwegian broodfish and the significantly different G4 subtype circulating in the Faroese smolt farms (Figure 4).

### 3.8. HPR0 Variants in Faroese Broodfish and Smolt Are Identical to Variants Circulating in the Marine Environment

The phylogenetic analysis of HPR0 variants detected between 2005 and 2014 in 134 individual salmon from 75 different marine fish cohorts revealed 25 unique HPR0 variants, all clustering in G2 and/or G4 (Appendix A, Appendix A). Most of the HPR0 variants detected in Faroese broodfish and smolt (both kept in land-based closed containment systems throughout production) were identical to the HPR0 variants detected in the marine environment (Appendix A). Of note, the detection of specific HPR0 variants in the marine environment preceded the detection of the matching HPR0 variant in broodfish or smolt in all of the cases, sometimes by several years (Appendix A), suggesting that the marine environment could be the source of HPR0 infections both in broodfish and smolt. Our G2- and G4-specific real-time RT-PCR revealed that 46% of the tested marine fish were co-infected with G2 and G4, whereas 34% were infected with G4 only and 19% were infected with G2 only (Figure 5c). Furthermore, we observed an increase in frequency of the G4 over time in the marine environment (Figure 5c, Appendix A).

### 3.9. Identical ISAV-HPR0 Variants May Persist in Smolt Farms over Several Years

Phylogenetic analyses of the HPR0 sequences obtained from the smolt farms indicated that specific HPR0 variants persist in the farm environment over several years. In smolt farm SII, identical G4 variants were observed over a three-year period (G4c in Figure 4), and in smolt farm SVI the same G4 variant was present three years after its first detection in 2012 (G4b in Figure 4). For a statistical evaluation of this observation, a Poisson regression model was used. The univariable model showed more similarity between sequences within a hatchery than between sequences from different hatcheries (*p* < 0.001, data not shown). When including both the hatchery and distance in time, the observed difference between the sequences increased over time (Table 3).

The 7-year predicted sequence difference within and between hatcheries further supported the observation of the increasing sequence difference between hatcheries over time (Table 4).

## 4. Discussion

A better understanding of ISAV-HPR0 epidemiology and infection dynamics is of major importance to underpin efficient biosecurity management in Atlantic salmon aquaculture. Here, we demonstrate a high and transient prevalence of the detection of HPR0 in land-based broodfish and in RAS-operated smolt farms on the Faeroe Islands, comparable to our previous findings in marine-reared Atlantic salmon [4]. Our comprehensive epidemiological and phylogenetic analysis of isolates from the three production stages of Atlantic salmon (i.e., broodfish, freshwater smolt and marine salmon) found no evidence for the transmission of HPR0 from broodfish to their progeny, strongly arguing against vertical transmission being the main route for the transmission of HPR0. On the contrary, our findings suggest that HPR0 is mainly transmitted horizontally from the marine environment to the land-based broodfish and smolt farms, as the detection of any one HPR0 variant in the marine environment preceded its detection in broodfish or smolt by several years. Although we found evidence suggestive of specific HPR0 variants being established as “house-strains” in the RAS-operated smolt-farms, new HPR0 variants also appear to be occasionally introduced from the marine environment.

### 4.1. HPR0 Transmission Pathways

In general, a virus may be transmitted by horizontal or vertical routes, or both. The question of whether ISAV is vertically transmitted has been, and still is, controversial [8]. Whereas horizontal transmission has been documented to be the main transmission pathway of virulent HPRΔ within and between marine-farmed Atlantic salmon fish groups [1,16,17,20,29,34], vertical transmission has been suggested to be the main transmission route for non-virulent ISAV-HPR0 [16,17,25]. During the present study period, ova and fry from HPR0-positive broodfish were sold to the Faroese freshwater smolt farms, hatched, and put into production. If the vertical transmission of HPR0 was a frequent phenomenon, one would expect to observe a close genetic relationship between the HPR0 variants detected in broodfish at stripping and those detected in progenies hatched from the same eggs. However, our comprehensive phylogenetic analysis does not support this model of transmission. Apart from a co-infection of HPR0 in the G2 and G4 subgroups in Faroese broodfish stripped in 2008, all of the HPR0 detected in both the Faroese and Icelandic broodfish belonged to G2. By contrast, the significantly different G4 subgroup was the predominant genotype detected in all five Faroese smolt farms, with only a single G2 variant detected in smolt farm V. Notably, this variant was detected as a co-infection with G4 early in 2008. Hence, the smolt harboring the G2 variant did not hatch from eggs stripped in late 2008. Based on our findings in 2008, i.e., that both broodfish and smolts can be co-infected with G2 and G4, we could not exclude the possibility that G4 existed as a low-frequency minority variant in the Faroese broodfish stripped in 2010 and 2011. Similarly, we could not exclude the possibility that G2 existed as a low-frequency minority variant in the smolt farms where we exclusively detected G4 between 2009 and 2014. However, we consider this unlikely, as the highly sensitive G2- and G4-specific assays, developed in this study did not detect G2 among the 234 HPR0-positive smolt sampled from 2009 to 2014, or G4 among the 149 HPR0-positive Faroese broodfish sampled in 2010 and 2011 (Figure 5). These findings were further supported by the next-generation sequencing of the *HE* gene of selected samples from both broodfish and smolts.

Several other observations argue against the vertical transmission of HPR0. First, the G2 and G4 variants isolated from the Faroese marine environment possess most of the phylogenetic diversity, and the Faroese smolt and broodfish variants are included in their phylogeny (Appendix A). This finding supports the supposition that the freshwater smolt and broodfish variants originated from the marine environment. Second, the two Icelandic broodfish farms were infected by genetically distinct HPR0 variants in the G2 subgroup, even though both farms received smolt only from the same hatchery. Moreover, the few cases of HPR0 reported so far in Norwegian broodfish were all within the G1 or G3 subgroups [16,17,26]. However, despite the extensive import of ova from both countries since 2003, and intensive surveillance for ISAV since 2005 [4,14], none of the HPR0 variants detected in Icelandic or Norwegian broodfish have ever been detected in any of the production stages in the Faroe Islands. In fact, whereas smolt farm SIV received ova exclusively from Norwegian broodfish between 2010 and 2013, only the Faroese G4 variant was detected in this smolt farm. Moreover, while smolt farm SIII received ova from the same sources as the other smolt farms, no HPR0 was detected on this farm throughout the study period. Taken together, these data strongly suggest that vertical transmission was not the main route for the introduction of HPR0 to the six Faroese smolt farms included in the present study. On the contrary, the geographical clustering of various HPR0 variants emphasizes the importance of horizontal transmission as a driving force of HPR0’s spread within and between the three production stages of Atlantic salmon in the Faroe Islands. From a managemental point of view, the appropriate disinfection of eggs—as implemented by the Faroese, Icelandic and Norwegian Broodfish companies—is important to eliminate HPR0 (or other pathogens) contamination on eggs in order to prevent subsequent transfer to progeny [35].

### 4.2. HPR0 House-Strains in Smolt Farms

The six smolt farms included in the present study were operated by RAS technology and continuous production. No year class separation or scheduled fallowing were implemented. Thus, a key question was whether HPR0, once introduced to a smolt farm, could persist in the environment over time. In order to examine this alternative, we used two approaches. First, we performed a phylogenetic analysis of the HE sequences from all of the smolt farms. The phylogenetic tree revealed that identical HPR0 variants were identified on the same farm over several years (Figure 4). In order to further elaborate on this, we next examined the HE sequences using a Poisson regression model. First, we found that the sequences within a smolt farm were more similar than between the farms (Table 3). Next, when including distance in time into the model, the sequence difference between the different smolt farms increased over time (Table 4). Based on these statistical observations, we propose that specific HPR0 variants were established as “house strains”, similarly to IPNV [36]. Despite the transient nature of HPR0 infections, it is possible that productive infection at low prevalence may pass from fish group to fish group, and may maintain the virus in the population between regular waves of high-prevalence infection. Another possibility is that HPR0 persists in a hitherto unknown niche in the RAS environment, from which new waves of infections with the same virus variant spread to new fish groups. The influenza virus has been shown to survive in feces, in lake sediments, and in surface water for up to one year in specific environmental conditions [37,38,39]. As ISAV belongs to the same virus family as the influenza virus and has a similar physical structure, it seems plausible that viable ISAV could survive for an extended time in biofilters or biofilms. Nevertheless, this needs to be tested experimentally or followed in longitudinal field studies, like this one. The emergence of house strains as a feature of ISAV-HPR0 infections in smolt farms also underpins the importance of closed containment systems and year class separation to prevent infections and diseases during industrial animal production [40]. This practice is well established in both poultry and pig production, and is considered to be essential for sustainable production systems.

### 4.3. Introduction of New HPR0 Variants to the RAS Smolt Farms and Broodfish Farm

In addition to the persisting HPR0 variants, occasionally, new variants appear to be introduced to smolt farms. Similarly, the broodfish cohorts seem to be infected by new HPR0 variants. What is the source of these HPR0 infections? We have previously shown that smolt may carry HPR0 when transferred to the sea [14]. However, contrary to what would be expected if these variants were predominantly transmitted by the transfer of smolt to the sea, the detection of individual HPR0 variants at marine sites preceded their detection in smolt farms, sometimes by several years (Appendix A). While these findings by no means preclude the transmission of HPR0 from land-based smolt farms to marine sites, they suggest a transmission model in which the infection of smolt farms predominantly originates from the marine sites. HPR0 primarily infects the epithelial cells of the gills and skin of Atlantic salmon [28]. Considering the high stocking density and number of individuals (>1 million) at a marine farming site, and considering that up to 100% of these individuals can be transiently infected with HPR0 [4], it is reasonable to expect the shedding of high numbers of HPR0-infected epithelial cells and mucus into the sea-surface microlayer [41]. The microlayer can be aerosolised into sea spray or sea foam, which is the main vector for the transport of bacteria and viruses across the air–sea interface [41]. Several studies have shown that marine aerosols large enough to contain viruses can be transported for hundreds of kilometres [40], facilitated by aerosolised organic aggregates [42]. Notably, the six smolt farms and the Broodfish farm on the Faroe Islands are located close to the coastline, and marine farms and may be exposed to HPR0-contaminated sea spray or sea foam, particularly during winter storms. And despite they smolt and Broodfish farms being closed containment systems and the intake water being filtered and UV-treated, the disinfection may not be 100% efficient. Furthermore, the degassing of CO_2_ in RAS requires the intake of large amounts of fresh air. These ventilators could therefore act as an entry-point for aerosols from sea spray potentially contaminated with pathogenic bacteria and viruses. Thus, strict biosecurity management strategies such as closed containment systems for smolt and broodfish production, and the efficient disinfection of intake water and air may reduce the risk of introducing ISAV-HPR0 or other pathogens from the marine environment.

### 4.4. Viral Fitness of HPR0 in G2 vs. G4

Surprisingly, our analysis revealed that the three production stages were dominated by HPR0 of different subgroups, i.e., G2 in broodfish supplied with seawater, G4 in freshwater smolt farms, and G2 and/or G4 in marine fish cohorts (Figure 5). Furthermore, we observed the increasing prevalence of G4 relative to G2 over time at marine production sites (Appendix A). Moreover, both of the recent (in 2014 and 2016) detections of virulent HPRΔ in Faroese aquaculture belonged to the G4 subgroup ([14], Christiansen DH, pers. comm.).

The G4 variant was first described as a HPRΔ from a mild ISA outbreak in Nova Scotia in 1999 [43]. Over the following 15 years, the G4 variant dominated HPR0 detections in Canada until the first HPR0 of the North American (NA) genogroup was found in 2012 [27]. Contrary to this, virulent HPRΔ variants were dominated by the NA genogroup. Subsequently, HPR0 G4 was detected in Scotland [44] and in the Faroe Island [4]. The G4 variant has not been detected in Norwegian aquaculture, but was recently reported from wild caught Atlantic salmon [17,24].

We cannot exclude the possibility that the different infection patterns of G2 and G4 are caused by stochastic infection dynamics. However, it seems unlikely that all five HPR0-positive smolt farms by chance were infected by G4 only when G2 and G4 co-circulate in the marine environment. So far, very little is known about how the considerable genetic diversity among ISAV genogroups and subgroups influences viral fitness in different environmental niches. Several studies have identified the existence of discrete genetic groupings and little genetic linkage between marine VHSV isolates and freshwater VHSV isolates [45,46], demonstrating that the different isolates have adapted to their specific environments [46]. It is therefore possible that the noticeable differences between the distribution of the G2 and G4 subgroups in our study is also related to different adaptations to the different production environments, or to specific development stages of Atlantic salmon. This interesting question should be addressed by future studies that map the genetic difference between the G2 and G4 subgroups at the whole-genome level, and link them to function and viral replication under different environmental conditions.

### 4.5. Conclusions

We here demonstrated that HPR0 causes a prevalent and transient infection in each of the three production stages of Atlantic salmon in the Faroe Islands. Our comprehensive genetic analysis demonstrated that there is little or no genetic link between the HPR0 found in broodfish and their progeny in the Faroese smolt farms, thus suggesting that HPR0 is not transmitted vertically. Contrarily, we found a close genetic link between HPR0 in salmon from the marine environment and in the smolt farms, as well as between the marine salmon and broodfish. Thus, our results argue for horizontal transmission being the main pathway for the dissemination of HPR0 within and between the three production stages of Atlantic salmon in the Faroe Islands, as depicted in Figure 6.

From a management point of view, the pattern of HPR0 infection with transient bouts of infection highlights the challenges associated with screening for HPR0 in each of the production stages. Thus, the documentation of the absence of HPR0 will be resource-demanding, will require frequent samplings in all of the epidemiological units, and will still be associated with a certain level of uncertainty.

Although all of the Faroese marine fish cohorts apparently experience a transient infection of HPR0 [4], a new HPRΔ was detected for the first time in 2014, almost 10 years after the devastating ISA epidemic [14]. This variant was linked genetically and epidemiologically to HPR0 detected in a smolt farm [14]. Thus, the transition to virulence seems to be a rare event, at least in marine farms where practical management strategies are adopted, as in the Faroes [4]. This risk is probably being further reduced by the production of large smolt where marine production time is markedly reduced. We speculate, on the other hand, on whether the increased production time of large smolt in RAS farms with continuous production and circulating HPR0 house strains might increase the risk of the maintenance, adaptation and evolutionary transition of HPR0 to HPRΔ. This risk is most likely low, but the consequences of moving HPRΔ smolts to several marine farms could potentially be high.

## Figures and Tables

**Figure 1 viruses-13-02428-f001:**
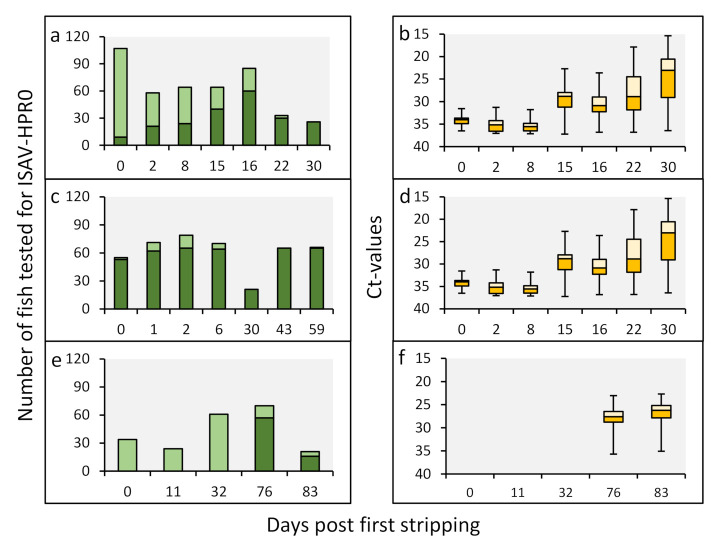
ISAV-HPR0 viral infection dynamics in the gills of Faroese Atlantic salmon broodfish. Throughout the stripping period, from September to January in 2008/09 (**a**,**b**), 2010/11 (**c**,**d**) and 2011/12 (**e**,**f**), the viral prevalence (**a**,**c**,**e**) and viral load (**b**,**d**,**f**) were measured by real-time RT-PCR, with cutoff values of the cycle threshold (Ct) ≥ 37, i.e., all of the samples with at Ct < 37 were considered to be positive. The prevalence is presented as the total number of HPR0-positive fish (dark green bar) plus total number of HPR0-negative fish (light green bar) at each of the stripping time points. The corresponding viral load is presented as a box blot (orange bars) at each of the stripping time points. The midhinge corresponds to the median Ct value. The lower and upper hinges correspond to the first and third quartiles. The lower and upper whiskers extend from the hinge to the smallest and largest values.

**Figure 2 viruses-13-02428-f002:**
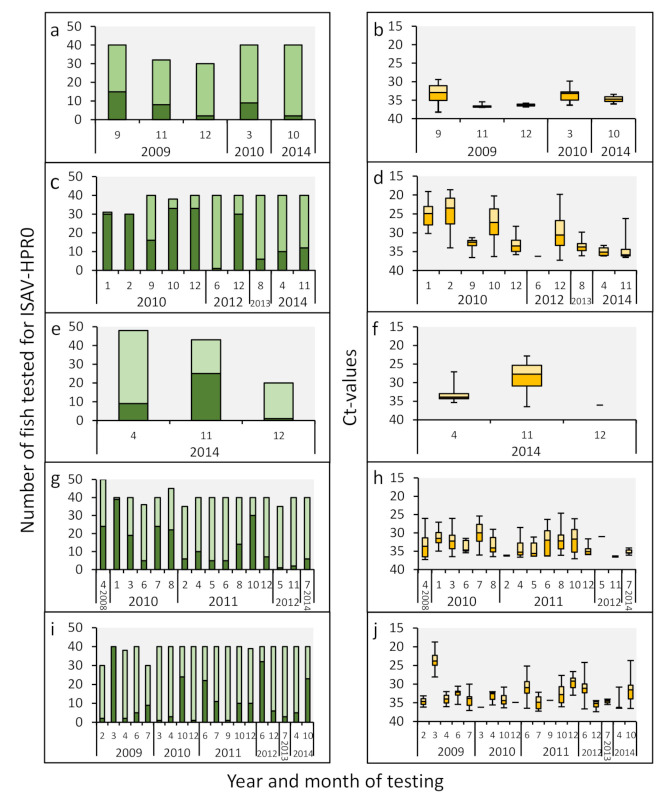
ISAV-HPR0 viral infection dynamics in the gills of Atlantic salmon smolt. For the study period from 2008 to 2014, the viral prevalence and load from smolt farm SI is presented in (**a**,**b**), smolt farm SII is presented in (**c**,**d**), smolt farm SIV is presented in (**e**,**f**), smolt farm SV is presented in (**g**,**h**), and smolt farm SVI is presented in (**i**,**j**). Smolt farm SIII did not test ISAV positive throughout the study period. The viral prevalence (**a,c,e,g,i**) and viral load (**b,d,f,h,j**) were measured and presented as outlined in Figure 1. Only sampling time points with HPR0-positive fish are included. The monthly HPR0 infection dynamics from each of the six smolt farms are presented in Appendix A.

**Figure 3 viruses-13-02428-f003:**
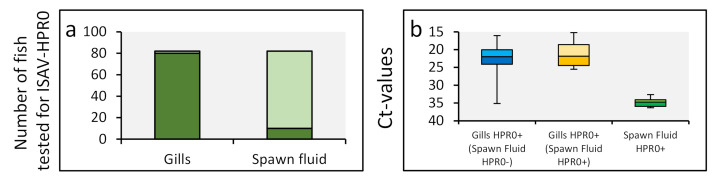
ISAV-HPR0 infection dynamics of the gills and spawn of Atlantic salmon brood fish. The HPR0 viral prevalence (**a**) and load (**b**) in the gills and spawn fluid of 82 Atlantic salmon broodfish stripped 30- and 59-days post first stripping in 2010 (Figure 1c) are presented. The viral prevalence and load were measured and presented as outlined in Figure 1. The blue box blot includes Ct-values from the 72 salmon of which the gills tested HPR0 positive and the corresponding spawn fluid tested negative. The orange and green box blots include the Ct values of the 10 fish of which the gills (orange box blot) and corresponding spawn fluid (green box blot) were HPR0 positive.

**Figure 4 viruses-13-02428-f004:**
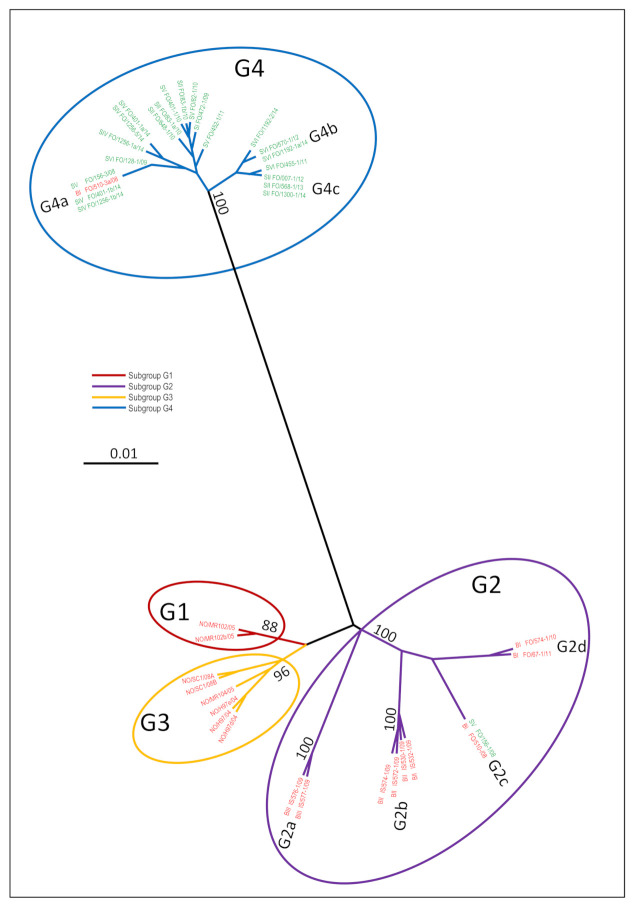
Phylogenetic radial tree showing the relationship between all of the 31 unique nonvirulent ISAV-HPR0 variants from Norwegian (NO), Icelandic (IS) and Faroese (FO) broodfish (red variant ID) and Faroese smolt (green variant ID). Included in the three are the 15 HPR0 variants (14 in G4 and one in G2) of the 21 Faroese smolt cohorts which tested positive between 2008 and 2014, the four variants (one in G4 and three in G2) of all three Faroese Broodfish cohorts, the six variants of the two Icelandic broodfish cohorts, and the eight unique and publicly available Norwegian Broodfish variants (see further details in Appendix A). The phylogenetic analysis was performed on 1154 nucleotides (nt 61 to 1214 relative to the start codon) of the haemagglutinin esterase gene. The phylogenetic relationship among the HPR0 variants was inferred using a maximum-likelihood analysis within the CLC Main Workbench 8.1 (Qiagen, Hilden, Germany). The branch length reflects the genetic distance. Bar: 0.01 substitutions per nucleotide side. Significant bootstrap values for the major subgroups were transferred to the unrooted tree derived from the original data.

**Figure 5 viruses-13-02428-f005:**
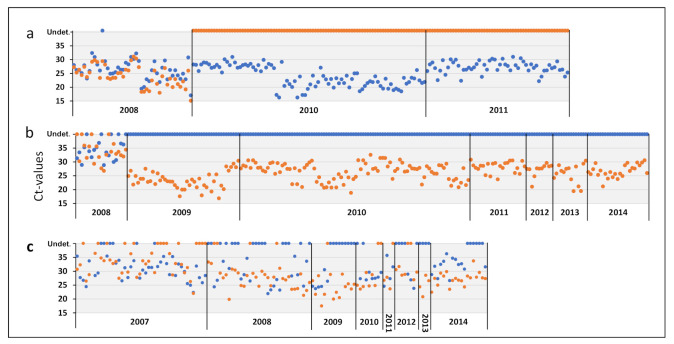
Infection dynamics of the G2 and G4 ISAV-HPR0 subgroups in Faroese Atlantic salmon broodfish (**a**), Faroese freshwater smolts (**b**), and Faroese Atlantic salmon marine grow-out sites (**c**). Each dot represents the Ct-values of individual fish, as demonstrated by the highly specific and sensitive G2 (blue dots) and G4 (orange dots) real-time RT-qPCR assays. The dots represent the Ct-values for each of the 195 HPR0-positive broodfish (representing the three broodfish cohorts) tested in 2008, 2010 and 2011; the 234 HPR0-positive smolts (representing 21 smolt cohorts) from the five smolt farms (SI, SII, SIV, SV and SVI) tested from 2008 to 2014; and the 139 HPR0-positive production fish (representing 75 production cohorts) from the 25 marine farming sites collected from 2007 to 2014. Included are only fish where the Ct-values of G2 and/or G4 were below 32. Whereas all of the 46 HPR0 positive broodfish tested at stripping in 2008 were co-infected with G2 and G4, all of the broodfish tested in 2010 (91 fish) and 2011 (58 fish) were infected with G2 only (**a**). All of the 213 smolts tested between 2009 and 2014 were infected with G4 only (**b**). Only the 21 smolts from smolt farm V tested at stripping in 2008 were co-infected with G2 and G4. In the marine environment, 64 of the 139 fish (46%) were co-infected with G2 and G4, whereas 26 (19%) and 49 (35%) were infected with either G2 or G4, respectively, throughout the study period from 2007 to 2014 (**c**). Undet.: undetected.

**Figure 6 viruses-13-02428-f006:**
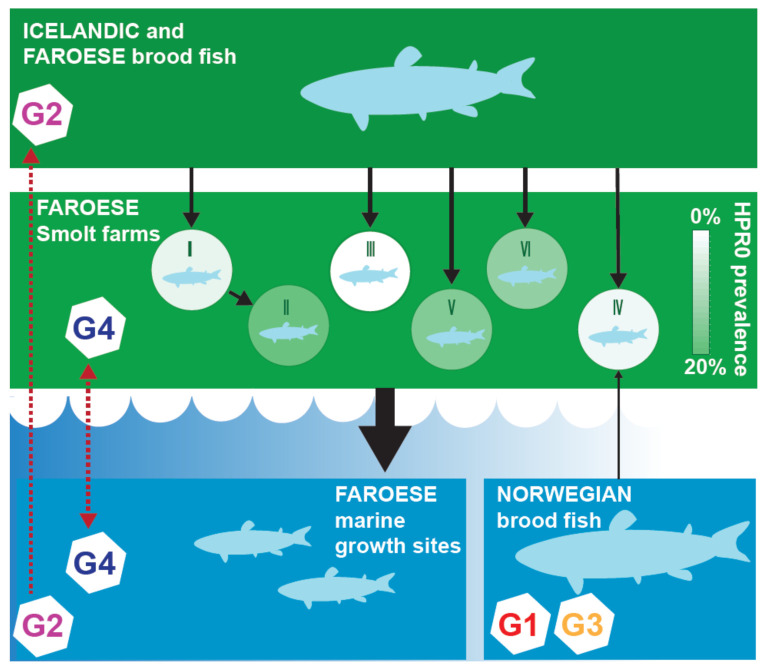
Carton showing the different production stages of Atlantic salmon from broodfish via ova to smolt to marine fish (black arrows) in the Faroe Islands, and the infection dynamics of the G2 and G4 HPR0 subgroups (red arrows) between the three production stages. Whereas broodfish delivering ova to the Faroese smolt farms were infected with G1, G2 or G3, only G4 was detected in the smolt farms, arguing against vertical transmission as the main pathway.

**Table 1 viruses-13-02428-t001:** Annual prevalence of ISAV-HPR0 in the three production stages of Atlantic salmon: broodfish, freshwater smolt and marine farmed salmon. The total number (n) of Atlantic salmon gills screened for ISAV and the number (n) and percentage (%) which tested ISAV positive by real-time RT-PCR throughout the study period from 2007 to 2014 are included. Included are samples from the only Faroese and the two Icelandic land-based broodfish farms delivering ova to the six Faroese freshwater smolt farms and the 25 Faroese marine farming sites.

	Faroese Broodfish at Stripping			Icelandic Broodfish at Stripping			Faroese Freshwater Smolt			Faroese Marine Farmed Salmon			Total		
	Total	HPR0 Positive		Total	HPR0 Positive		Total	HPR0 Positive		Total	HPR0 Positive		Total	HPR0 Positive	
Year	n	n	%	n	n	%	n	n	%	n	n	%	n	n	%
2007	256	9	3.5	*			639	0	0.0	5387	811	15.0	6026	811	13.0
2008	443	210	47.4	*			788	39	4.9	9066	1100	12.1	10,201	1349	13.2
2009	50	0	0.0	2374	455	19.2	1954	83	4.2	8847	852	9.6	13,223	1392	10.5
2010	427	395	92.5	4502	183	4.1	1792	288	16.1	6686	235	3.5	13,407	1101	8.2
2011	210	73	34.8	6120	110	1.8	2129	131	6.2	5404	159	2.9	13884	473	3.4
2012	263	1	0.4	2320	6	0.3	395	72	18.2	3560	135	3.8	6547	214	3.3
2013	65	0	0.0	2425	117	4.8	360	15	4.2	2018	81	4.0	4873	214	4.0
2014	121	76	62.8	1272	3	0.2	498	87	17.5	2474	374	15.1	4365	541	12.4
Total	1835	764	41.6	19,013	874	4.6	8555	715	8.4	43,442	3747	8.6	72,526	6095	8.4

*: no sampling.

**Table 2 viruses-13-02428-t002:** Annual prevalence of ISAV-HPR0 in freshwater smolt farms. The total number (n) of Atlantic salmon gills screened for ISAV and the number (n) and percentage (%) which tested ISAV positive by real-time RT-PCR throughout the study period from 2007 to 2014 at the six Faroese freshwater smolt farms (I to VI) are included.

	Smolt Farms																	
	SI			SII			SIII			SIV			SV			SVI		
	Total	HPR0 Positive		Total	HPR0 Positive		Total	HPR0 Positive		Total	HPR0 Positive		Total	HPR0 Positive		Total	HPR0 Positive	
Year	n	n	%	n	n	%	n	N	%	n	n	%	n	n	%	n	n	%
2007	80	0	0.0	*			240	0	0.0	80	0	0.0	80	0	0.0	159	0	0.0
2008	80	0	0.0	60	0	0.0	208	0	0.0	160	0	0.0	120	39	32.5	160	0	0.0
2009	332	25	7.5	269	0	0.0	357	0	0.0	341	0	0.0	307	0	0.0	348	58	16.7
2010	309	9	2.9	219	142	64.8	284	0	0.0	359	0	0.0	361	108	29.9	260	29	11.2
2011	320	0	0.0	316	0	0.0	380	0	0.0	359	0	0.0	395	77	19.5	359	54	15.0
2012	80	0	0.0	80	31	38.8	40	0	0.0	40	0	0.0	75	3	4.0	80	38	47.5
2013	40	0	0.0	40	6	15.0	80	0	0.0	120	0	0.0	40	6	15.0	40	3	7.5
2014	74	2	2.7	80	22	27.5	69	0	0.0	111	35	31.5	84	0	0.0	80	28	35.0
Total	1315	36	2.7	1064	201	18.9	1658	0	0.0	1570	35	2.2	1462	233	15.9	1486	210	14.1

*: no sampling.

**Table 3 viruses-13-02428-t003:** Results from the multivariable Poisson regression model for the hatchery HPR0 sequences, with the origin (same or different hatchery) and distance (days between sampling points) as explanatory variables.

	Estimate	Std.error	*p*
Intercept	1.307	0.061	<0.001
Same hatchery	−0.569	0.105	<0.001
Days difference	0.000115	0.00005	0.014

**Table 4 viruses-13-02428-t004:** The expected 7-year HPR0 sequence difference within and between hatcheries. For each year, the mean difference is shown, together with the 2.5 to 97.5 confidence interval in brackets.

	Year							
	0	1	2	3	4	5	6	7
Same hatchery	2.1 (0.5)	2.2 (0.5)	2.3 (0.6)	2.4 (0.6)	2.5 (0.6)	2.6 (0.6)	2.7 (0.6)	2.8 (0.6)
Different hatchery	3.7 (1.8)	3.9 (1.8)	4.0 (1.8)	4.2 (1.9)	4.4 (1.9)	4.6 (1.9)	4.8 (1.9)	5.0 (1.1)

## Data Availability

All of the relevant data are within the paper and its supporting information files or deposited in the GeneBank database.

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
