# Peer review of "No Evidence of the Vertical Transmission of Non-Virulent Infectious Salmon Anaemia Virus (ISAV-HPR0) in Farmed Atlantic Salmon"

_viruses, 2021, doi:10.3390/v13122428_

Round 1

Reviewer 1 Report

This article “No evidence of vertical transmission of non-virulent infectious 2 salmon anaemia virus (ISAV-HPR0) in farmed Atlantic salmon” is packed with important information and observations, though my feeling is that it could use even more. My recommendations are listed below.

Abstract, line 17: this is a sweeping statement that may or may not be true. I suggest adding the word ‘putative’ ‘in front of progenitor. And removing the word ‘all’.

Line 47: Add virulent to this sentence: All known ‘virulent’ HPRΔ variants ….  This is because segment 5 mutations do not always accompany segment 6 HPR mutations. Either that or here define the term HPRΔ to mean the co-occurrence of both (to distinguish HPRΔ from the term HPR deleted).

Line 52: HPR0 as wild-type is the predominant theory. However, it is not yet demonstrated that the deletions/insertions occur only in the stated direction (HPR0 to HPRΔ). The ubiquitous presence of HPR0 questions whether it most commonly presents as fodder or resolution, or entirely distinct, to outbreaks. I’d recommend that these types of statements are posed as theory rather than fact. For example, change ‘represents’ to ‘is considered’.

Line 77: Studies ‘have’.

Line 99: How were these study periods selected? What is the reason the two groups were not evaluated over the same periods (2007-2014)?

Line 101: What portion of broodfish farms (egg source) are represented in this selection? Providing rationale for the exclusions (2 smolt farms and some number of broodfish farms) will help the reader rule in or out concerns re selection bias.

Line 105: Are these farms cleaned and disinfected between cohorts? Are eggs surface disinfected before use? Are the broodstock and smolt farms located in different geographic regions/watersheds? Is effluent water treated? 

Line 110: ‘based on the HPR0 findings’ … This is first mention of HPR0 findings in Icelandic farms. It might be helpful to further describe.

Line 116: Was start feeding flow-through water also UV-treated? If so, were UV settings sufficient to handle that water volume (for ISAV)?  

Line 158: How were the 22 samples selected?

Figures 1 and 2: It would be good to also see sampling events without positive fish as this helps display the periodicity (and variation) in occurrence. Maybe these are already included? If so, remove the last line in the Figure 1 description. Also – the Figure description seems to have flipped the meaning of dark and light green bars (at least relative to the discussion in the text). The text makes it seem that dark green bars (not light green) represent HPR0 positives.

Figure 4: It would be interesting to see a phylogenetic analysis using segment 5. Was any work done on this front?

Figure 4. Also – when you chose one of each variant for phylogenetic analysis, how did you select among those that occurred at different farms? Some of the inferences made using this chart (e.g., that smolt farms were predominantly G4 and brood farms predominantly G2) are only possible if you’ve also looked at the source of like variants not here included. It would be good for the reader to be able to track these conclusions. On further reading, it looks like you’ve covered this in later sections. Consider switching the order?

Missing: It would be very interesting to see the variation in sequence distribution. Do like sequences cluster in time and space? And are certain sequences more common/persistent? Does this vary between smolt and broodstock farms? This would help inform the degree to which HE sequencing sequencing data may or may not support epidemiologic inference. On further reading I see this discussed in the conclusion, though it seems it should be presented in results as well.

Figure 5: 2008 marks an interesting transition, from co-infection in both smolt and brood facilities to single (and different) genogroup infection in each. Can you speculate on what might have happened at the end of the 2008 cycle that could have contributed to this transition?

Figure 5: Similarly, how might this chart change if you mapped results by watershed rather than facility or brood/smolt type?

Figure 5: Consider changing ‘40’ to ‘undetected’ in the vertical axis label. It’s easy to overlook this intention and misinterpret the graph.

Section 3.8. So broodfish farms are sourced by marine-reared fish? Do all or some marine farms source broodstock facilities? Are they tested first, and are movement decisions influenced by test results? Perhaps this information could be presented in the earlier sections describing farming practices.  

Lines starting 485. This is where a discussion of egg surface disinfection practices could be valuable.

Line 558. Spelling error – resent versus recent.

Line 579. Whether variants identified by HE sequence analysis may or may not represent what one might find through whole genome analysis deserves some cautionary discussion.

Line 593. It also would be interesting to discuss whether and how often HPRΔ occurred in this region during the study period, and whether its detection preceded or followed its closest HPR0 match? Given the high and persistent backdrop of HPR0, what does this add to our understanding of the relationship between HPR0 and HPRΔ? Do we really need to worry about HPR0 - is high-expense screening for HPR0 an appropriate use of limited resources??

Author Response

Dear reviewer 1

First I would like to thank you for all the constrictive comments which I think has improwed the manuscript significantly.

Attached please find my point by point comments

Reviewer 2 Report

The manuscript by Debes Christiansen and colleagues present the analysis of an extensive dataset and testing of different compartments of Atlantic salmon production to shed light on trasmission pattern of HPR0 variants of ISAV. THe manuscript is generally well written, the amount and quality of data high, and overall it appears as an excellent piece of work.

A few comments for consideration to elaborate on the manuscript.

1- There is no mentioning whatsoever on disinfection procedures of eggs  from broodstock station to smolt station . this should be reported and also help elaborating in the discussion

2- it is mentioned in fig S1 and in hte text NGS analysis, but this does not seem thoroughly presented in the text, and would probably deserve so (?)

3- what happened to Faroese broodstock farm in 2009 (ref. fig 5) there was no detection of ISAV HPR0 . how does the author elaborate on this finding?

4- table 1 and 2 refer to "annual presence of ISAV" is this ISAV HPR0 or the findigns have been typed? 

5- table S3 report monthly prevalence of ISAV HPR0. Would the author use these data to elaborate a recommendation on when to sample for specific surviellance purposes of HPR0?

Author Response

Dear reviewer 2

First I would like to thank you for the constructive comments which I think has improved the manuscript

Attached please find my comments point by point
